# Perceptions about and reasons for participation in research bronchoscopy in Uganda: A qualitative analysis

**David Kaawa-Mafigiri** [1,2☯] *, **Mary Nsereko** [2☯], **Michael Odie** [2☯], **John L. Johnson** [2,3☯]

**1** School of Social Sciences, Makerere University Kampala, Kampala, Uganda, **2** Uganda-Case Western Reserve University Research Collaboration, Kampala, Uganda, **3** Department of Medicine, Case Western Reserve University School of Medicine and University Hospitals Cleveland Medical Center, Cleveland, Ohio, United States of America

☯ These authors contributed equally to this work.
* mafigiridk@yahoo.com

**Data Availability Statement:** The minimal, de-identified data set underlying the results of this manuscript is located on a local server at the Uganda-Case Western Reserve University

## Abstract

This study sought to assess perceptions towards and reasons for participation in research bronchoscopy studies in a high TB burden urban setting. Additionally, the study aimed to identify areas of pre- and post-procedural concern among healthy adults approached to participate in research bronchoscopy. A cross sectional qualitative study was undertaken at the Uganda-Case Western Reserve University Collaboration Tuberculosis Research Project Clinic at Mulago National Referral Hospital in Kampala, Uganda. In-depth interviews were conducted with participants at their pre-bronchoscopy visit (n = 17) and after they had undergone bronchoscopy (n = 23) to examine their perceptions and experiences with the procedure. Following consent, all interviews were audio recorded and later transcribed and typed in MS WORD. Local language interviews were translated into English by the social science interviewers. Qualitative analysis was performed manually following an inductive and emergent approach typical in thematic analysis. This study was approved by the Makerere University School of Social Sciences Research Ethics Committee (MAKSS REC 09.18.220) and registered with the Uganda National Council for Science and Technology (UNCST SS4785). Overall willingness to participate in bronchoscopy was high as many participants viewed the study as primarily a means of getting free health checks and determining their health status. Notably, despite extensive face to face counseling for this study coupled with the fact that our participants had been involved in prior research at the site, therapeutic misconception still played a pivotal role in willingness to participate in research bronchoscopy. Therapeutic misconception has important ethical and research implications in clinical research, which requires strategies to tackle it, even among a pool of potential participants who are knowledgeable about a disease or clinical care procedures. Continuous awareness and knowledge building about the difference between being a trial participant and therapeutic misconception must become a mainstay in trials to improve the process of informed consent for future research bronchoscopy studies.

Collaboration Tuberculosis Research Project Clinic's data center. It is now available through email (data@mucwru.or.ug) upon reasonable request, with permission and notification of the Makerere University School of Social Sciences Research Ethics Committee (MAKSS REC). Any potentially identifying patient information has been fully anonymized.

**Funding:** The parent study was funded by grants and contracts UO1AI115642, RO1AI124348 and 75N93019C00071 from the U.S. National Institutes of Health. The funders had no role in study design, data collection and analysis, decision to publish, or preparation of the manuscript.

**Competing interests:** The authors have declared that no competing interests exist.

## Introduction

In Uganda, bronchoscopy with and without bronchoalveolar lavage has predominantly been used by specialist pulmonary physicians for diagnosis and clinical care [1]. Situations where it has been useful include the diagnosis of smear negative tuberculosis (TB) and other opportunistic lung infections such as pneumocystis pneumonia in HIV infected individuals, the diagnosis of lung cancer, and removal of aspirated foreign bodies [1]. The use of bronchoscopy as a research tool has mainly been limited to patients with clinical lung disease [1]. However, recent studies of healthy individuals exposed to patients with TB have begun to examine human protective immune responses against TB [2, 3]. Little is known about the reasons that motivate otherwise healthy individuals to undergo or decline research bronchoscopy.

Previous exploration of reasons for participation in research bronchoscopy studies among volunteers in Africa [2]. Europe and the United States revealed personal benefit, including health assessment, perceived future access to health care when ill, and remuneration as factors influencing willingness to undergo bronchoscopy. Studies also suggest that altruistic reasons including the desire to help current or future patients with lung diseases are common motivations for participation. On the other hand, inconvenience associated with research and fear of procedures, have been considered as barriers to participation [2, 4–6]. Regardless of the motivation or barriers to participation, it is vital to understand the factors underlying these choices to improve quality of research participation and engagement. Understanding factors that influence choices about participation in research bronchoscopy will inform researchers about areas of concern and causes of anxiety in potential research participants. This knowledge may also contribute to ensuring informed consent for participation in research bronchoscopy. Attaining informed consent requires an understanding of the design and content of materials used to deliver and share information (making informed consent more relevant to the participants) and working with participants before, during, and after research bronchoscopy. The Uganda-Case Western Reserve University Research Collaboration (UCRC) is a long-standing research collaboration between Makerere University in Kampala, Uganda and Case Western Reserve University, Cleveland, U.S. The Collaboration began performing research bronchoscopy with bronchoalveolar lavage in healthy adults in Uganda in December 2017 to obtain cells from the lungs for studies of human immunity against TB in household contacts exposed to patients with active TB. These studies examine the immunopathogenesis of TB infection and progression to active TB in HIV-infected and–uninfected individuals. To improve participant knowledge about research bronchoscopy, assure informed consent and allay participant anxiety, the investigators involved social scientists to interview participants and collect information about their perceptions and experience with research bronchoscopy.

### Objectives

This study sought to assess perceptions towards and reasons for participation in research bronchoscopy studies by healthy adults in a high TB burden urban setting. Additionally, the study aimed to identify areas of pre- and post-procedural concern among healthy adults approached to participate in research bronchoscopy.

## Materials and methods

### Study setting

A cross sectional qualitative study was undertaken in 2019 at the UCRC Research Project Clinic at Mulago National Referral Hospital in Kampala, Uganda. UCRC is a research collaboration between Makerere University in Kampala and Case Western Reserve University in

Cleveland, US with over 30 years of experience conducting clinical and other health sciences research in TB, HIV and other non-infectious diseases. The authors have been conducting research with the UCRC for various lengths of time in the past ranging between 15–32 years. They oversaw this current study's procedures including participant recruitment, interviewer selection and data collection. In-depth interviews were conducted with participants at their pre-bronchoscopy visit (n = 17) and after they had undergone bronchoscopy (n = 23) to examine their perceptions and experiences with the procedure. The interviews were conducted by two trained social scientists with previous experience in health services research including previous social science studies conducted at UCRC. The interviewers spoke both English and Luganda, the most widely spoken local language in the study area. The interviewers were supervised for quality control by both the Clinic Director and the Principal Investigator who is an Anthropologist and senior social science researcher at UCRC.

## Study procedures

Adult participants in an ongoing cross-sectional observational study of the immunopathogenesis of TB in individuals who did or did not become infected with M. tuberculosis after being exposed to a person in their household with active pulmonary TB who had given their permission to be contacted about additional TB research studies were contacted about participating in a study of blood and lung immune responses against TB that included research bronchoscopy with bronchoalveolar lavage and a research blood collection. After giving informed consent, volunteers underwent a history and physical examination, a chest radiograph to exclude active TB or other clinical lung disease, and complete blood count, chemistry, and coagulation testing. Full details are described elsewhere [7]. Participants were scheduled for a pre-bronchoscopy visit three days before the procedure, a bronchoscopy visit, and a post bronchoscopy visit three days after the procedure.

For this qualitative study, two social scientists independent of the clinic staff who provided care to research participants of the parent bronchoscopy study approached the participants at the pre bronchoscopy visit to seek their interest in the study. Willing participants were consented, enrolled and interviewed. The consent included a second follow up interview at the post bronchoscopy visit three days after the procedure.

Participants were interviewed in the language of their choice, either Luganda or English using an interview guide. Staff at the TB clinic worked with the social scientists to schedule interviews during both the pre and post bronchoscopy visits. Interviews were held with individual participants in a comfortable private room that ensured privacy and confidentiality.

All interviews were audio recorded and later transcribed and typed in MS WORD. Local language interviews were translated into English by the social science interviewers.

Logistical constraints and interview scheduling led to some participants not having both pre and post bronchoscopy interviews. Of the 6 participants who had only a pre-bronchoscopy interview, 4 were ineligible for and did not undergo the bronchoscopy procedure consequent to out of range screening laboratory test results. The participants who were scheduled and available for a post bronchoscopy visit but had missed the pre-bronchoscopy interview were retrospectively interviewed about their pre bronchoscopy perceptions.

## Analysis

Qualitative analysis was performed manually following an inductive and emergent approach typical in thematic analysis [8]. Data was coded and organized through indexing or measurement device to assign values to the text. Drawing from the broader reflexive thematic analytical strategy [9, 10], we relied on the 'codebook approach' to develop a codebook (see attached

supplementary material). Two research assistants independently coded data with a third (study researcher) acting to determine consensus during discussions as a team. The codebook was then pilot tested on 10 interviews (5 from pre- and 5 from post-broncho participants) before finalizing it. The coding process was iterative and ongoing and used some a priori codes selected beforehand and in vivo codes that emerged during analysis [8, 11]. Initially 2 coders worked independently with one single researcher synthesizing the codes into broader themes. Selected specific themes were refined further to develop a final list of themes and theme definition. Some codes that were set a priori focused on what can be changed, reasons for participating in a bronchoscopy study, and advice to other people who might participate in bronchoscopy. These a priori codes were influenced by adaptations from the health belief model [12, 13] which posits that individuals will seek care (alter their behavior) once the perceived risk to their health (susceptibility to a condition like TB) supersedes other alternatives, including 'riding it out' (if a condition is not severe), self-medication or treatment (usually when they cannot afford associated costs).

The pre-bronchoscopy interviews focused on experiences if this was their first time to participate in this kind of study, at what age they started participating in this kind of research, their relationship with any person that had TB, and their feelings towards bronchoscopy. An in depth review of the literature on the acceptability and perceptions about research bronchoscopy was also conducted and helped to generate some codes. Our literature review included recent studies on this topic including studies done in low and middle income studies and tuberculosis.

### Ethical approvals

This study was approved by the Makerere University School of Social Sciences Research Ethics Committee (MAKSS REC 09.18.220) and registered with the Uganda National Council for Science and Technology (UNCST SS4785). All participants gave written informed consent for study participation.

### Results

Of the 31 participants interviewed for this study, 17 (55%) were male and 28 (90%) were aged less than 30 years. Nine (29%) of the participants had both a pre and post bronchoscopy interview (Table 1).

**Table 1. Demographic characteristics of participants.**

| Characteristic | f | % |
| --- | --- | --- |
| | (n = 31) | |
| **Sex** | | |
| Male | 17 | 55 |
| Female | 14 | 45 |
| **Age** | | |
| 21–30 years | 28 | 90 |
| Above 30 years | 3 | 10 |
| **KII Conducted** | | |
| Pre Bronchoscopy only | 8 | 26 |
| Post Bronchoscopy only | 14 | 45 |
| Both Pre & Post Bronchoscopy | 9 | 29 |

Mean age 25.7 ± 6.2; Min 19; Max 45; Median 25.

The major themes that emerged for willingness to participate in research bronchoscopy as derived from reasons for participation were prior experience of TB having witnessed the suffering and healing process of loved ones as they underwent care from the UCRC; a need to 'know their health status' given the range of tests and other care procedures associated or anticipated with being a participant on the study; and therapeutic misconception about gaining treatment despite being healthy adults without TB. An important theme was the willingness to participate for altruistic reasons indicating the appreciation of how research may help generate important knowledge for future health care technology and services.

## Reasons for participation

Many participants (n = 13) reported that their motivation to undergo research bronchoscopy was due to the care a loved member in their social network had received previously at the health facility. For some, their parents, other relatives, or friends had received free medical care services for TB treatment.

*The other reason why I participated in this research bronchoscopy was that my husband was treated from here at no cost and he got better. . . Aaaah I was thanking them because I was grateful for what they did for us (Female, Post Bronchoscopy)*

*What gave me the courage? I think of course I had that fear but when I sat back and remembered when my dad was ill, we literally got free treatment and these guys were with him until he got better and he had to go back to work. Free treatment, and that medicine was really, really expensive you should go to [know what is in] private hospitals, we tried to do research outside and it was expensive. It is an intensive kind of treatment that does not stop in one day or two. You can even take a full year taking tabs, but then these people aided my dad and they did not stop only on my dad but they also aided all of us (Female, Post Bronchoscopy)*

To them, participating in the study was seen as 'giving back' to the society (altruism). Study participants emphasized that when their loved ones were sick, the UCRC clinic provided them with free medical care services that would otherwise not have been affordable.

*Yes, I did it because, I had a brother who had TB way back and the organization helped to provide treatment for him and we as a family we were all joined into [recruited for follow-up] the organization for treatment because the TB drugs were expensive. So, when they explained to me that there is what they want to do with my help for the good of the TB patients, I said why shouldn't I help, if at all they came and provided treatment for the person they did not know without giving them any money for the whole year giving him all the necessary treatment. So, I was like why shouldn't I do this small thing such that another person benefits. That is why I decided to do it (Male, Post-bronchoscopy)*

Another important reason for willingness to undergo research bronchoscopy was the anticipation that participants would receive care and treatment just in case they had TB or other related illness that may be discovered. Whereas these participants had clearly been informed that they were not TB infected, and that one of the criteria for joining the study was being a healthy adult, they appeared to perceive that by undergoing bronchoscopy, if found to have TB they would be treated. One participant narrated that:

*Both parents were victims of TB, so the doctors were wondering why we did not get infected, so because of that reason they started to 'diagnose' us with an offer that in case they find us*

*infected they would provide the treatment and I said to myself I cannot miss such a chance/ opportunity."(Male Post Bronchoscopy)*

Other participants noted:

*Why I wish to participate, I will get to know whether my health is okay, even to know my viral load "Obutafali" and even to inform my other family members those that we have been receiving treatment with that I am normal or I have the TB infection like my uncle, my sibling and others so that they get strong. Because the other week my uncle requested me to come to help him know the truth whether I am infected or not, this will help them to get stronger as me being their example" (Female, Pre Bronchoscopy)*

*The reason is that I want to know whether I am infected with TB or not because the outside may not show yet when inside is examined it might be seen. What I expect is to get results telling me whether I am sick inside or not; I hope to get those results (Female, Pre Bronchoscopy)*

We note that our study population comprised of household contacts of an index TB patient and as such participants likely had some experience with seeing the severity of TB among other household members. For instance, many had reported seeing their loved ones suffer from effects of TB, possibly influencing their perception that if they participated in the study, they too might benefit from those trial procedures. Other participants, who knew that they were healthy at the time they volunteered for the research bronchoscopy study expressed gratitude for TB care given to their family members, loved ones and others in their local community by the UCRC TB Project Clinic. As such they were motivated to participate in support of their loved ones.

*The reason why I did accept to have bronchoscopy done was because my father had TB, he came here and was treated and got cured at no cost. So I did this because if my father was here as a TB patient and he was treated by Mulago [Hospital name housing the UCRC] Tuberculosis why don't I do the same if I am called upon? The other reason is because I also used to come here for treatment although I was not infected with TB. Therefore I did this for my parent (Male, Post Bronchoscopy)*

Overall, many people reported participating due to therapeutic misconception. Therapeutic misconception among research subjects occurs when they do not distinguish between clinical research and ordinary treatment, and therefore inaccurately attribute therapeutic intent to research procedures in the clinical studies they participate in [14–16].

Five respondents agreed to participate in the bronchoscopy study because the health workers (counselors, nurses and doctors) had reportedly informed them that they were trying to discover a drug that helps in curing TB. Participants stressed that this was an important stage as it will help reduce the number of people dying from TB in Uganda and other parts of the world.

The desire to 'know their health statuses' was another important reason given for participation in bronchoscopy. Given the fact that the study involved a number of screening procedures including laboratory tests, participants were keen to know their health status. This kept them active, throughout the whole study. And as medical test results were always returned and interpreted to the study participants, this encouraged them to proceed to a next level and as such complete the bronchoscopy procedure.

*I like this proverb that. . . (How do you call it); prevention is better than cure. So, when I come, they check me, see what is inside my lungs, the oxygen in my blood, the pressure. So that*

*helps me to know my health status and I was curious about it so I had to come in (accept to participate in bronchoscopy).* (*Female, Post-bronchoscopy*)

Some participants (n = 3) pointed out that their motivation to participate in research bronchoscopy was love of adventure and desire to make history. They wanted to be remembered for having played a big role in the discovery of TB drugs in Uganda. Their participation was attributed to the fact they want to be affiliated with a group of researchers working towards the discovery of a TB drug.

*Okay I am a learned person and I know the importance of research. I would like to be part of the research, big research that will help people. Sometimes you can do research and it is just for news but this is not for news. It will help people once it becomes successful. So, I am one person who can help among all so I cannot just be there yet I can help to save as many people as this. So, the most important thing is that this study will not be harmful to my body, that is what matters most.* (*Female, Post-bronchoscopy*)

*I was very happy [to participate] because I thought they want me to stay safe from diseases* (*Male, Post-bronchoscopy*)

## Perception of bronchoscopy

The majority of participants (n = 12), who had undergone bronchoscopy considered it largely satisfactory and acceptable. They pointed out that they did not have any major discomfort or problems during and after the procedure. They were willing to participate in future research including studies involving research bronchoscopy. Some respondents (n = 5) reported experiencing transient discomfort during bronchoscopy, especially when inserting the bronchoscope through the nose and into the throat and lungs. Even those who had not yet undergone/experienced bronchoscopy had the same fear of discomfort associated with inserting the tube.

Some participants found it very hard to convince their relatives and friends that they were going to participate in a study involving an invasive procedure like bronchoscopy. Some reported leaving their homes to attend the clinic for bronchoscopy without informing their relatives or family members where they were going or why. One participant pointed out that her parents had threatened to denounce her as their daughter if she participated in the bronchoscopy study; however, this didn't stop her from participating in the study. She attributed her resolve to the confidence and trust she had gained from the medical doctors.

*I was informed by my parents that if I participate in this study, I cease being their daughter and they are not responsible for any complication that might result from the bronchoscopy participation. But the health workers had assured me that I will not have any complications so I escaped and participated.* (*Female, Post-bronchoscopy*)

This illustrates that, while some participants view bronchoscopy as acceptable, there remains negative perceptions towards bronchoscopy in the community as they think it might lead to serious health consequences.

Some participants (n = 3) perceived the bronchoscopy and the required pre- and post-procedure visits to be time consuming which interfered with their work schedules. They suggested that the process needs to be shortened so that it did not become disruptive to their work schedules.

*There was a challenge because there was where I was working from and as an employee, I was missing work for some days and they threatened to dismiss me. (Female, Post-broncho)*

A number of participants mentioned the desire to help the researchers achieve their goal and help their fellow man in the future

"*...My primary goal is when the research has been successful. I don't care how long it will take but I want to know that I was helpful*" (Male, Pre-broncho)

"*First and foremost I want to help them with the research they are carrying because there are many people with TB. ...so if I can be their research sample to get what they want that is the one reason I am taking part in the study*" (Female, Pre-bronchoscopy)

"*I feel this is the way of rewarding and giving back to the community and regarding the reasons why they are doing this research*" (Female, Pre-bronchoscopy)

## Discussion

This study aimed at determining the perceptions and reasons for participating in a research bronchoscopy study among healthy adults who were household contacts of TB patients in Kampala. Overall willingness to participate in bronchoscopy was high as many participants viewed the study as primarily a means of getting free health checks and determining their health status. There appears to be an overlap among participants who may be willing to participate in bronchoscopy research as a way to learn more about their health status and gain other treatment benefits associated with being a research participant, particularly in a context of low social economic status like our study setting. For instance, at the UCRC, numerous tests including blood tests, chest radiographs, and sputum examinations for TB, and research bronchoscopy are performed on research participants and as such some participants are motivated to participate in a study due to the perceived treatment related to the research. However, all participants in our research studies receive intensive, focused consent and health education from experienced counselors, nurses and physicians that they are participating in a research study and not a TB treatment study. All participants had multiple opportunities to ask questions about the study and have their questions answered. We therefore note that the plausible therapeutic element in this case was mainly a misconception on the part of the otherwise healthy adults. This therapeutic misconception was also one of the major findings of a study done in Malawi [2]. In the context of constrained public health services at all levels in LMIC coupled with the poor socioeconomic status of participants, this finding is not surprising. What is notable is that despite adequate counseling for this study coupled with the fact that our participants had been involved in prior research at the site, therapeutic misconception still played a significant role in willingness to participate in research bronchoscopy. Therapeutic misconception has been shown to have serious ethical and research implications in trials [14–16]. Future research bronchoscopy studies should consider strategies to decrease therapeutic misconception, even among a pool of potential participants who are knowledgeable about a disease or clinical care procedures.

Another commonly mentioned reason for participation was the valued clinical and social care the respondents and other members of their households had previously received at the study clinic. As mentioned earlier, the UCRC research and associated health care services has been ongoing for over 30 years in this study context. Moreover, the authors as well as the study staff have also had two or more decades of familiarity with research in the study area. We recognize that our interpretations of participants' reasons for willingness to participate in research

bronchoscopy may be influenced by our position as longstanding researchers in this area. For instance, it is possible that our understanding and interpretation of some of the narratives about willingness to participate may be borne out of our bias as longstanding researchers in the area. However, we are confident in our methodological rigor such as ensuring that we used independent, trained social science research assistants to conduct the interviews. This was done in part to improve on internal validity of the interview process. We still acknowledge that for participants, there is a possibility that the association with the UCRC broader research and treatment agenda may have overridden the efforts to ensure they answer honestly about their perceptions of research bronchoscopy.

Our study is potentially influenced by adaptations of the health belief model [12, 13] which states that individuals will seek care once they perceived risk to their health supersedes other alternatives, mainly 'riding it out' self-medication or treatment. Because some of the participants indicated that their willingness was influenced in part due to their prior experience seeing loved ones being treated well for TB, and the gravity of TB, it is possible that they felt a need to also participate in efforts associated with TB control. Research bronchoscopy in this case would have been perceived in that lens, and so participants may have selected to undergo the procedure having perceived TB as so severe not to engage with care. These findings illustrate the need for continuous/ongoing awareness and knowledge building about the difference between being a research participant and therapeutic misconception. This knowledge gap has implications for improving the process of informed consent for future research bronchoscopy studies. The fact that there can be blurred lines between an individual's motivation to participate in a study due to anticipated health benefits, even when not clearly stated, and the misperception about gaining therapy despite repeated concerted effort to clarify that participation is based on being healthy and there is no direct treatment benefit to participants, further emphasizes the need to continuously provide awareness and knowledge building to potential research participants.

Our study has several limitations. Our results may not be generalizable to other settings as our participants all had prior research experience at the study site and this may have influenced their acceptability for the research bronchoscopy procedure. Moreover the study participants reside in the catchment area of the UCRC research clinic activities which in some cases means they may have encountered care directly or indirectly observed care of index TB cases in their households. Indeed, for this specific study as pointed out earlier in the methods, the participants were household contacts of a known TB case. However, our findings underscore the need for continuous education and information about research bronchoscopy even among populations that may have significant exposure to research environments. As such the process of informed consent for instance may benefit from having up-dated and ongoing assessment of reasons for and willingness to participate in research bronchoscopy.

## Supporting information

**S1 File. Pre broncho questionnaire.**
(PDF)

**S2 File. Post broncho questionnaire.**
(PDF)

## Acknowledgments

We thank the participants who generously gave us their time and attention to complete this study.

## Author Contributions

**Conceptualization:** David Kaawa-Mafigiri, Mary Nsereko, John L. Johnson.

**Data curation:** David Kaawa-Mafigiri, Michael Odie, John L. Johnson.

**Formal analysis:** David Kaawa-Mafigiri, Mary Nsereko, Michael Odie.

**Funding acquisition:** John L. Johnson.

**Investigation:** Mary Nsereko.

**Methodology:** David Kaawa-Mafigiri, Mary Nsereko, Michael Odie.

**Project administration:** David Kaawa-Mafigiri, Mary Nsereko, Michael Odie, John L. Johnson.

**Supervision:** David Kaawa-Mafigiri, Michael Odie, John L. Johnson.

**Writing – original draft:** David Kaawa-Mafigiri, Mary Nsereko.

**Writing – review & editing:** David Kaawa-Mafigiri, Mary Nsereko, Michael Odie, John L. Johnson.

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
