## [Decision Letter · Decision Letter 0]

9 Mar 2023

PONE-D-23-01054Perceptions about and reasons for participation in research bronchoscopy in Uganda: a qualitative analysisPLOS ONE

Dear Dr. Mafigiri,

Thank you for submitting your manuscript to PLOS ONE. After careful consideration, we feel that it has merit but does not fully meet PLOS ONE’s publication criteria as it currently stands. Therefore, we invite you to submit a revised version of the manuscript that addresses the points raised during the review process.

Your manuscript has been comprehensively  assessed by one reviewer, and their full report is available below. The reviewer appreciated the research question of your study and the importance of your findings, but they also raised several points about methodology reporting and data analysis, as well as the discussion of the results. Please carefully address all concerns raised. 

We are issuing a decision on your manuscript at this point to prevent further delays in the evaluation of your manuscript. Please be aware that the editor who handles your revised manuscript might find it necessary to invite additional reviewers to assess this work once the revised manuscript is submitted. However, we will aim to proceed on the basis of this single review if possible. 

We look forward to receiving your revised manuscript.

Kind regards,

Dario Ummarino, PhD

Senior Editor

PLOS ONE

Journal Requirements:

“The parent study was funded by grants and contracts UO1AI115642, RO1AI124348 and 75N93019C00071 from the U.S. National Institutes of Health.”

6. Please include your tables as part of your main manuscript and remove the individual files. Please note that supplementary tables (should remain/ be uploaded) as separate "supporting information" files.

Reviewers' comments:

Reviewer's Responses to Questions

**Comments to the Author**

1. Is the manuscript technically sound, and do the data support the conclusions?

Reviewer #1: Partly

2. Has the statistical analysis been performed appropriately and rigorously? 

Reviewer #1: N/A

3. Have the authors made all data underlying the findings in their manuscript fully available?

Reviewer #1: No

4. Is the manuscript presented in an intelligible fashion and written in standard English?

Reviewer #1: Yes

5. Review Comments to the Author

Reviewer #1: The background and concept for this qualitative exploration of research participants' motives for undoing research bronchoscopy is excellent - the kind of qualitative research which should accompany all clinical research programmes. The engagement of locally embedded social scientists strengthens both the rationale and the methodological rigour of the work, and I think the findings are valuable and worthy of sharing.

The early part of the paper is very convincing, and the background exploration is succinct and well written. The key aspects of the paper that I think need developing are the methods and the results sections - which will have a knock-on effect on the discussion section.

From the information provided this study was well established and delivered, and I have no reason to doubt either the quality of breadth of the data collected by the research team. However, there are aspects missing which i think are essential:

1. A reflexivity statement from the authors explaining who they are, their background, and how this shapes both the questions asked and the likely way in which the interviews generated data. This needs contextualising in Uganda and the way people engage with hierarchy and research activities. I think the key to interpreting qualitative data is in understanding any potential biases or confounding factors in the data - why might the research participants chosen give the particular answers they did to these researchers at this time and in this context? Might an alternative study find different accounts (for example if interviewing them in their home setting, or in a group?) and why might these accounts vary? Would a different research team find different results because of the nature of the interviewers?

2. The details of the analysis are too thin for me at the moment. All the right work has been done, but I need to better understand the a priori codes and where they came from, how the literature was reviewed, what research team experience went into them etc. The a priori codes are so important as they expose the underlying logic model of the research team, the results they might 'expect' to find and the existing themes within the literature. The literature review methods is also essential - was this systematic or narrative, and how might any biases in it inform these a priori codes? Following this, I think it's important to see a bit more of the code book, the themes that began and then emerged, and how they were grouped and/or abandoned to come up with the final themes.

3. Developing this point, I think a clear understanding of the analysis technique would help. there are now som many variations on reflexive thematic analysis and it can be useful to understand which variant was chosen and why. If theme generation is they key, these themes need to be clearly listed and defended (sometimes a rich infographic is best for these to try to show how they interact with each other and with any overarching or meta themes).

4. These themes then need to pull through more strongly into the results and discussion. At the moment, the statements are quite weak 'some participants responded...' and even where the number is quoted such as n=12 this does not really convey how strongly these points were made in the data. I also felt the need to really understand more what people said (I suspect no respondent actually used the words 'therapeutic misconception' so it is really important to see how this term emerged from their lived experiences). The quotes used are excellent but a little sparse. Could there be more in accompanying data if space does not permit longer quotes in the body text?

5. The core notion of therapeutic misconception I felt needed more unpacking. To the newcomer, this sounds like people thinking they are going to get treatment which they are not. However it seems to overlap with people using research participation to understand more about their own health, which is not seemingly the same thing. Having a load of blood tests and a bronco definitely will tell you something about your health, even if not therapeutic - I wonder where these boundaries lie. Did some participants really think they were going to get better as a result of the branch study? Presumably in the ethics for the study, and health issue which is identified must then also be treated or referred - so in essence maybe there is a valid therapeutic element to participating? Is this then a misconception or simply a different perspective?

Trying to bring these points together, I feel as though there needs to be some revision to help the interested reader get closer to the lived experience of the participants, and then understand the inferences drawn by the research team to synthesise these into generalisable concepts. This mostly involves a more detailed account of the data creation and analysis to show how the data was obtained and processed to create these insights, but once done will prompt a sightly revised discussion section.

For all this, I reiterate that I think the paper is an excellent concept and am sure the data obtained is of a high quality - I just to need to be shown more of it!

6. PLOS authors have the option to publish the peer review history of their article (what does this mean?). If published, this will include your full peer review and any attached files.

Reviewer #1: No

---

## [Author Response · Author response to Decision Letter 0]

3 May 2023

PONE-D-23-01054

Perceptions about and reasons for participation in research bronchoscopy in Uganda: a qualitative analysis

PLOS ONE

Response to Reviewer Comments

Review Comments to the Author

Reviewer #1: 

Comment:

1. A reflexivity statement from the authors explaining who they are, their background, and how this shapes both the questions asked and the likely way in which the interviews generated data. This needs contextualising in Uganda and the way people engage with hierarchy and research activities. I think the key to interpreting qualitative data is in understanding any potential biases or confounding factors in the data - why might the research participants chosen give the particular answers they did to these researchers at this time and in this context? Might an alternative study find different accounts (for example if interviewing them in their home setting, or in a group?) and why might these accounts vary? Would a different research team find different results because of the nature of the interviewers?

Response:

We have expanded discussion of the issues of reflexivity and potential bias to the findings due to the research context in the introduction (lines 98-100) and the study limitations section (lines 365-371). The Uganda-CWRU Research Collaboration’s (UCRC) TB Project Clinic has been in operation in this area for over 30 years. As such the chances that the study participants or their household members have previously heard about or has prior contact with health care at the clinic are high. All of the participants were household contacts of index TB patients who received TB treatment from the UCRC clinic. They may have experienced some exposure to the UCRC research and clinic operations. However we strongly believe that given the trained social scientists and study procedures to gain consent that the responses were not biased to a point where they may negatively impact the conclusions. 

Comment:

2. The details of the analysis are too thin for me at the moment. All the right work has been done, but I need to better understand the a priori codes and where they came from, how the literature was reviewed, what research team experience went into them etc. The a priori codes are so important as they expose the underlying logic model of the research team, the results they might 'expect' to find and the existing themes within the literature. The literature review methods is also essential - was this systematic or narrative, and how might any biases in it inform these a priori codes? Following this, I think it's important to see a bit more of the code book, the themes that began and then emerged, and how they were grouped and/or abandoned to come up with the final themes.

Response:

Thank you for the comments on better understanding the coding process. We conducted a detailed review of the limited medical literature about perceptions and acceptability of research bronchoscopy in clinical research in low and middle income countries including correspondence and sharing of experience with investigators in Malawi and Norway who have worked in this area. We have added additional description (lines 156-160) explaining how we selected some codes a priori – from literature on the subject which we reviewed specifically from prior knowledge. We have attached our code books with this resubmission to illustrate how we reached the final themes. An extract of the codebooks could be included as supplementary material in the publication. We feel that the comment and explanations have strengthened the analysis section.

Comment:

3. Developing this point, I think a clear understanding of the analysis technique would help. there are now so many variations on reflexive thematic analysis and it can be useful to understand which variant was chosen and why. If theme generation is they key, these themes need to be clearly listed and defended (sometimes a rich infographic is best for these to try to show how they interact with each other and with any overarching or meta themes).

Response:

We have expand discussion on the analytical process (lines 143-146) as described above and included additional references (see below and references 9-10 on line 144) supporting and explaining the approach in more detail. We agree that theme generation was an important part of the analysis and has an influence on how we draw our conclusions. 

The additional references are:

Braun, V., Clarke, V.: Thematic analysis. In: Cooper, H., Camic, P.M., Long, D.L., Panter, A.T., Rindskopf, D., Sher, K.J. (eds.) APA Handbook of Research Methods in Psychology, Research Designs, vol. 2, pp. 57–71. American Psychological Association, Washington (2012)

Braun, V., Clarke, V.: Reflecting on reflexive thematic analysis. Qual. Res. Sport Exerc. Health 11(4), 589–597 (2019). https:// doi. org/ 10. 1080/ 21596 76X. 2019. 16288 06

Comment:

4. These themes then need to pull through more strongly into the results and discussion. At the moment, the statements are quite weak 'some participants responded...' and even where the number is quoted such as n=12 this does not really convey how strongly these points were made in the data. I also felt the need to really understand more what people said (I suspect no respondent actually used the words 'therapeutic misconception' so it is really important to see how this term emerged from their lived experiences). The quotes used are excellent but a little sparse. Could there be more in accompanying data if space does not permit longer quotes in the body text?

Response:

We have revised the results and discussion to strengthen them and discuss the lived experiences in more depth. We have added additional vignettes to the paper to provide more insight about therapeutic misconception. The discussion now includes the important point about blurred lines between actual, plausible intent to gain some health care benefits as the UCRC clinic and the misperception that undergoing bronchoscopy as part of research was likely to lead to TB treatment. The additional text can be found in the Results (lines 178-189; 202-239) and the Discussion (lines 329-340; 356-361; 365-371).

Comment:

5. The core notion of therapeutic misconception I felt needed more unpacking. To the newcomer, this sounds like people thinking they are going to get treatment which they are not. However it seems to overlap with people using research participation to understand more about their own health, which is not seemingly the same thing. Having a load of blood tests and a bronco definitely will tell you something about your health, even if not therapeutic - I wonder where these boundaries lie. Did some participants really think they were going to get better as a result of the branch study? Presumably in the ethics for the study, and health issue which is identified must then also be treated or referred - so in essence maybe there is a valid therapeutic element to participating? Is this then a misconception or simply a different perspective?

Trying to bring these points together, I feel as though there needs to be some revision to help the interested reader get closer to the lived experience of the participants, and then understand the inferences drawn by the research team to synthesise these into generalisable concepts. This mostly involves a more detailed account of the data creation and analysis to show how the data was obtained and processed to create these insights, but once done will prompt a slightly revised discussion section.

For all this, I reiterate that I think the paper is an excellent concept and am sure the data obtained is of a high quality - I just to need to be shown more of it!

Response:

We agree with both comments 4 and 5 above and have revised and expanded the Results and Discussion. The additional text discussing therapeutic misconception can be found in the Results (lines 178-189; 202-239) and Discussion (lines 329-340; 356-361; 365-371). We have now brought the reader closer to the lived experiences by adding more quotes from the data. We have also attached the codebooks which may be provided as supplementary material for the reader to further engage with the way we engaged with the lived experiences. 

Journal Requirements

Comment:

Response:

The manuscript now meets PLOS ONE's style requirements, including those for file naming.

Comment:

Response:

We have amended the funding statements provided in the first submission and added information that was originally stated in the acknowledgements section yet it should have appeared in the funding statement. The new funding statement reads as follows:

“The parent study was funded by grants and contracts UO1AI115642, RO1AI124348 and 75N93019C00071 from the U.S. National Institutes of Health. The funders had no role in study design, data collection and analysis, decision to publish, or preparation of the manuscript.”

Comment:

“The parent study was funded by grants and contracts UO1AI115642, RO1AI124348 and 75N93019C00071 from the U.S. National Institutes of Health.”

Response:

We have amended the Acknowledgement section by removing funding-related text from the manuscript (lines 378-380). The new acknowledgement text reads as follows:

“We thank the participants who generously gave us their time and attention to complete this study.”

Comment:

Response:

The minimal, de-identified data set underlying the results of this manuscript is located on a local server at the Uganda-Case Western Reserve University Collaboration Tuberculosis Research Project Clinic’s data center. It is now available through email (data@mucwru.or.ug) upon reasonable request, with permission and notification of the Makerere University School of Social Sciences Research Ethics Committee (MAKSS REC). Any potentially identifying patient information has been fully anonymized.

Comment:

Response:

The minimal, de-identified data set underlying the results of this manuscript is located on a local server at the Uganda-Case Western Reserve University Collaboration Tuberculosis Research Project Clinic’s data center. It is now available through email (data@mucwru.or.ug) upon reasonable request, with permission and notification of the Makerere University School of Social Sciences Research Ethics Committee (MAKSS REC). Any potentially identifying patient information has been fully anonymized.

Comment:

6. Please include your tables as part of your main manuscript and remove the individual files. Please note that supplementary tables (should remain/ be uploaded) as separate "supporting information" files.

Response:

We have included our table as part of the main manuscript (lines 170-171) and removed the individual file. We have included extracts of the codebook as separate "supporting information" files under supplementary material.

---

## [Decision Letter · Decision Letter 1]

19 Jun 2023

PONE-D-23-01054R1Perceptions about and reasons for participation in research bronchoscopy in Uganda: a qualitative analysisPLOS ONE

Dear Dr. Mafigiri,

Thank you for submitting your manuscript to PLOS ONE. After careful consideration, we feel that it has merit but does not fully meet PLOS ONE’s publication criteria as it currently stands. Therefore, we invite you to submit a revised version of the manuscript that addresses the points raised during the review process.

We look forward to receiving your revised manuscript.

Kind regards,

Daniel Semakula, M.D. MPH, PhD

Academic Editor

PLOS ONE

Journal Requirements:

Additional Editor Comments:

Would you please respond o the comments raised by the editors and please submit the full data compilation associated with this study?

Reviewers' comments:

Reviewer's Responses to Questions

**Comments to the Author**

1. If the authors have adequately addressed your comments raised in a previous round of review and you feel that this manuscript is now acceptable for publication, you may indicate that here to bypass the “Comments to the Author” section, enter your conflict of interest statement in the “Confidential to Editor” section, and submit your "Accept" recommendation.

Reviewer #1: (No Response)

Reviewer #2: (No Response)

2. Is the manuscript technically sound, and do the data support the conclusions?

Reviewer #1: Yes

Reviewer #2: Partly

3. Has the statistical analysis been performed appropriately and rigorously? 

Reviewer #1: N/A

Reviewer #2: I Don't Know

4. Have the authors made all data underlying the findings in their manuscript fully available?

Reviewer #1: Yes

Reviewer #2: No

5. Is the manuscript presented in an intelligible fashion and written in standard English?

Reviewer #1: Yes

Reviewer #2: Yes

6. Review Comments to the Author

Reviewer #1: Thanks for these careful amendments in response to my original review - they go a long way to addressing my questions. I have only minor comments in response.

1. 'A reflexivity statement from the authors'. I think my original point has been slightly missed. I have faith in the expertise of the authors and their study protocol, and do not question their conclusions. However all qualitative research has a degree of subjectivity and inherent bias, which I think needs more reflection and understanding when presenting the findings. Importantly, this does not weaken the conclusions, in my view. As the authors note, their collaboration has been running a long time and there will be a great degree of knowledge of it (as indeed comes across in the quotes). Given this, and the position of the researchers as affiliated to the programme, how might this have focussed the responses given by the participants? I would expect any person in any context to have a range of complex motivations for a given action (some implicit, some examined), underpinned by a biopsychosocial narrative about their health. What such narratives exist in Uganda? What might be the role of faith, health beliefs, and community pressure and which of these might not be volunteered to academic researchers affiliated to an established health outreach programme? The authors also started with a code book drawn from literature (as opposed to say a Community Engagement exercise) - how does this direct the interpretation of the data generated? I think it's important to identify a grounding in a given model of health, disease, and culture so that the findings are grounded in a given perspective. The few lines added do not quite address this for me and I am intrigued (for my own interest as well as academic rigour) to read what the author perceptions are as to what things they were told but also might not have been told, and why the respondents might have given that particular account at that particular time. Is it possible to see the prompting questions for the interviews?

2 & 3. 'The details of the analysis'. This is well addressed, but I still do not see a clear delineation of the final codes and themes identified at the completion of the rTA? If a thematic analysis has been perforrmed, can the authors say clearly what their final themes are and how they are linked?

4. 'These themes then need to pull through more strongly into the results and discussion'. The addition of more quotes really helps, thanks for this. I could still use some clearer signposting as to how the quotes chosen link directly to very clear thematic findings, but they very well articulate the broad discussion areas which are worked through.

5. 'The core notion of therapeutic misconception.' This was much better in this version, and I felt like I got the heart of it. The extra quotes again really helped. Annoyingly I could not see the attached codebooks on Editorial Manager which I'm sure is a software issue rather than the authors' fault!

In summary I think this remains a great paper. I still feel the need for more critical insight into the authors' views as to why they as a research team got this data, and what data they might have missed or chosen not to elicit. I'm still left with a slightly vague feeling as to the key themes the team feel they want to highlight, and how they feel these themes might be linked in a casual or narrative model. However these things are increasingly minor points, driven by a reviewer who is interested in both the subject matter and the methodology!

Reviewer #2: I am not too sure if I quite approve of this study. This is a research study done on healthy subjects and the authors have gone for an invasive procedure like bronchoscopy which could have been bypassed for non invasive or less invasive procedures maybe.The authors themselves have said there were lots of apprehensions regarding the procedure. Could they give more explicit details regarding whether they didn’t consider it necessary for a bystander to be accompanying the subject when they were doing an invasive procedure? If in case , there would have been complications in the event of bystanders not being there, how would they have dealt with it?How many subjects did not have bystanders accompanying them for the procedure? Was this point not raised by the IEC?This being a small study and going by what the authors say that it was done in a research sensitised area, the results cannot be generalised.

7. PLOS authors have the option to publish the peer review history of their article (what does this mean?). If published, this will include your full peer review and any attached files.

Reviewer #1: No

Reviewer #2: No

---

## [Author Response · Author response to Decision Letter 1]

2 Aug 2023

PONE-D-23-01054R1

Perceptions about and reasons for participation in research bronchoscopy in Uganda: a qualitative analysis

PLOS ONE

Response to Reviewer Comments

Review Comments to the Author

Reviewer #1: 

Comment:

Thanks for these careful amendments in response to my original review - they go a long way to addressing my questions. I have only minor comments in response.

1. 'A reflexivity statement from the authors'. I think my original point has been slightly missed. I have faith in the expertise of the authors and their study protocol, and do not question their conclusions. However all qualitative research has a degree of subjectivity and inherent bias, which I think needs more reflection and understanding when presenting the findings. Importantly, this does not weaken the conclusions, in my view. As the authors note, their collaboration has been running a long time and there will be a great degree of knowledge of it (as indeed comes across in the quotes). Given this, and the position of the researchers as affiliated to the programme, how might this have focussed the responses given by the participants? I would expect any person in any context to have a range of complex motivations for a given action (some implicit, some examined), underpinned by a biopsychosocial narrative about their health. What such narratives exist in Uganda? What might be the role of faith, health beliefs, and community pressure and which of these might not be volunteered to academic researchers affiliated to an established health outreach programme? The authors also started with a code book drawn from literature (as opposed to say a Community Engagement exercise) - how does this direct the interpretation of the data generated? I think it's important to identify a grounding in a given model of health, disease, and culture so that the findings are grounded in a given perspective. The few lines added do not quite address this for me and I am intrigued (for my own interest as well as academic rigour) to read what the author perceptions are as to what things they were told but also might not have been told, and why the respondents might have given that particular account at that particular time. Is it possible to see the prompting questions for the interviews?

Response:

Thank you for this additional clarification on the earlier comment. We appreciate the need to provide more insight and discussion about the issue of reflexivity. We note that in fact our position in the program may influence our interpretation of the participants’ views just as their responses may be influenced by the background context of their engagement with UCRC’s research and treatment agenda. As mentioned earlier, the UCRC research and associated health care services has been ongoing for over 30 years in this study context. Moreover, the authors as well as the study staff have also had two or more decades of familiarity with research in the study area. We recognize that our interpretations of participants’ reasons for willingness to participate in research bronchoscopy may be influenced by our position as longstanding researchers in this area. For instance, it is possible that our understanding and interpretation of some of the narratives about willingness to participate may be borne out of our bias as longstanding researchers in the area. However, we are confident in our methodological rigor such as ensuring that we used independent, trained social science research assistants to conduct the interviews. This was done in part to improve on internal validity of the interview process. We still acknowledge that for participants, there is a possibility that the association with the UCRC broader research and treatment agenda may have overridden the efforts to ensure they answer honestly about their perceptions of research bronchoscopy. 

Our study is potentially influenced by adaptations of the health belief model (HBM): 

Rosenstock IM, Strecher VJ, Becker MH. Social learning theory and the Health Belief Model. Health Educ Q. 1988 Summer;15(2):175-83. doi: 10.1177/109019818801500203. 

Abraham, C., & Sheeran, P. (2015). The health belief model. Predicting health behaviour: Research and practice with social cognition models, 2, 30-55). 

The HBM posits that individuals will seek care once the perceived risk to their health supersedes other alternatives, mainly ‘riding it out’ self-medication or treatment. Because some of the participants indicated that their willingness was influenced in part due to their prior experience seeing loved ones being treated well, and the gravity of TB, it is possible that they felt a need to also participate in efforts associated with TB control. Research bronchoscopy in this case would have been perceived in that lens, and so participants may have selected to undergo the procedure having perceived TB as so severe not to engage with care. Part of this explanation is now included in the manuscript on lines 156-160 in the analysis section and 377-384 in the discussion section.

We have attached the interview guide here for the reviewer to see the anticipated probes (prompting questions).

Comment:

2 & 3. 'The details of the analysis'. This is well addressed, but I still do not see a clear delineation of the final codes and themes identified at the completion of the rTA? If a thematic analysis has been perforrmed, can the authors say clearly what their final themes are and how they are linked?

Response:

We have added additional explanation to link the main themes to the codes (see lines 180-187 in the findings section). The major themes that emerged for willingness to participate in research bronchoscopy as derived from reasons for participation were prior experience of TB having witnessed the suffering and healing process of loved ones as they underwent treatment; a need to ‘know their health status’ given the range of tests and other care procedures associated or anticipated with being a participant on the study; and therapeutic misconception about gaining treatment despite being healthy adults without TB. An important theme was willingness to participate for altruistic reasons indicating an appreciation of how research may help generate important knowledge to improve care for future patients.

Comment:

4. 'These themes then need to pull through more strongly into the results and discussion'. The addition of more quotes really helps, thanks for this. I could still use some clearer signposting as to how the quotes chosen link directly to very clear thematic findings, but they very well articulate the broad discussion areas which are worked through.

Response:

We have added more clarity in the findings and discussion about the main themes as discussed above. Some additional subheadings and signposting has been added to link the findings to the discussion.

Comment:

5. 'The core notion of therapeutic misconception.' This was much better in this version, and I felt like I got the heart of it. The extra quotes again really helped. Annoyingly I could not see the attached codebooks on Editorial Manager which I'm sure is a software issue rather than the authors' fault!

Response:

Thank you for appreciating the additional clarity to explain the notion of therapeutic misconception. We have re-attached the codebooks and hope they will be visible/accessible.

Comment:

In summary I think this remains a great paper. I still feel the need for more critical insight into the authors' views as to why they as a research team got this data, and what data they might have missed or chosen not to elicit. I'm still left with a slightly vague feeling as to the key themes the team feel they want to highlight, and how they feel these themes might be linked in a casual or narrative model. However these things are increasingly minor points, driven by a reviewer who is interested in both the subject matter and the methodology!

Response:

Thank you for this helpful comment. We were motivated to undertake this study because of the growing use of bronchoscopy in both research and treatment contexts. But we were unaware what the community of participants in our catchment area appreciate about the procedure as used for research in healthy individuals. We needed to confirm some of the reasons for willingness to participate in an invasive procedure for research purposes when they are healthy adults without TB. All the information will help us to further improve the counseling and education sessions our investigators and staff have with research participants. We have discussed these points further in the discussion and added additional text to clarify these themes in the revised manuscript (see lines 363-384).

Reviewer #2: 

Comment:

I am not too sure if I quite approve of this study. This is a research study done on healthy subjects and the authors have gone for an invasive procedure like bronchoscopy which could have been bypassed for non invasive or less invasive procedures maybe.The authors themselves have said there were lots of apprehensions regarding the procedure. Could they give more explicit details regarding whether they didn’t consider it necessary for a bystander to be accompanying the subject when they were doing an invasive procedure? If in case, there would have been complications in the event of bystanders not being there, how would they have dealt with it? How many subjects did not have bystanders accompanying them for the procedure? Was this point not raised by the IEC? This being a small study and going by what the authors say that it was done in a research sensitised area, the results cannot be generalised.

Response:

Thank you for these important observations. Fiberoptic bronchoscopy with bronchoalveolar lavage is a widely used safe method that has been used to study lung diseases such as asthma, interstitial lung disease, sarcoidosis and tuberculosis over the past 40 years. Clinical research units in South Africa and Malawi also perform research bronchoscopy with bronchoalveolar lavage in many of their studies. The main mode of transmission of tuberculosis in humans is inhalation of infectious aerosol droplet nuclei containing small numbers of viable tubercle bacilli. Initial interactions with immune cells in the lung are likely to be important and can not be studied in blood samples. The main objective of the parent study for this substudy of participant acceptability and perceptions about research bronchoscopy in healthy adults in Uganda is to study immunological responses to TB infection in the lung and blood to understand host responses to TB infection and inform the development of safer and more effective TB vaccines, a major global public health need. The parent study is sponsored by the US National Institutes of Health and has undergone full scientific and ethical review and approval in Uganda and the U.S. Two local independent medical safety monitors, who are chest physicians, monitor the study. Adverse events have been infrequent and have consisted mainly of minor self-limited sore throat and cough.

Consent for research bronchoscopy is obtained by trained nurse counselors and the investigators. Participants also receive an IRB-approved information sheet about research bronchoscopy. Participant's have multiple opportunities to ask questions and have them answered before bronchoscopy. Participants are free to withdraw from the study at any time.

Participants undergo full medical evaluation before research bronchoscopy. Research bronchoscopy is done by experienced chest physicians assisted by trained nurses with an anesthetist on standby. Participants undergo monitoring of their their pulse, blood pressure and blood oxygen saturation during bronchoscopy and are observed by medical staff for one hour after the procedure and seen in follow-up 48 to 72 hours later.

Additional Editor Comments

Comment:

Would you please respond to the comments raised by the editors and please submit the full data compilation associated with this study?

Response:

We have responded to the comments raised by the editors and submitted full data compilation for this study.

---

## [Decision Letter · Decision Letter 2]

27 Sep 2023

PONE-D-23-01054R2Perceptions about and reasons for participation in research bronchoscopy in Uganda: a qualitative analysisPLOS ONE

Dear Dr. Mafigiri,

Thank you for submitting your manuscript to PLOS ONE. After careful consideration, we feel that it has merit but does not fully meet PLOS ONE’s publication criteria as it currently stands. Therefore, we invite you to submit a revised version of the manuscript that addresses the points raised during the review process.

Your revised manuscript has been re-reviewed by reviewer #2, whose comments you can see below. This reviewer is satisfied that their earlier concerns have been addressed. However, I note that although reviewer #1 requested a copy of the interview guide, it does not appear in your submission. Please could you upload a copy of your interview guide as a supporting information file?

We look forward to receiving your revised manuscript.

Kind regards,

Steve Zimmerman, PhD

Associate Editor, PLOS ONE

Journal Requirements:

Reviewers' comments:

Reviewer's Responses to Questions

**Comments to the Author**

1. If the authors have adequately addressed your comments raised in a previous round of review and you feel that this manuscript is now acceptable for publication, you may indicate that here to bypass the “Comments to the Author” section, enter your conflict of interest statement in the “Confidential to Editor” section, and submit your "Accept" recommendation.

Reviewer #2: All comments have been addressed

2. Is the manuscript technically sound, and do the data support the conclusions?

Reviewer #2: Yes

3. Has the statistical analysis been performed appropriately and rigorously? 

Reviewer #2: I Don't Know

4. Have the authors made all data underlying the findings in their manuscript fully available?

Reviewer #2: Yes

5. Is the manuscript presented in an intelligible fashion and written in standard English?

Reviewer #2: Yes

6. Review Comments to the Author

Reviewer #2: I think all comments have been addressed to satisfaction and it is a great paper. We perhaps need to motivate the authors to undertake more such studies so that we have a better insight into the disease.

7. PLOS authors have the option to publish the peer review history of their article (what does this mean?). If published, this will include your full peer review and any attached files.

Reviewer #2: No

---

## [Author Response · Author response to Decision Letter 2]

2 Oct 2023

PONE-D-23-01054R2

Perceptions about and reasons for participation in research bronchoscopy in Uganda: a qualitative analysis

PLOS ONE

Response to Reviewer Comments

Associate Editor’s comments: 

Comment:

Your revised manuscript has been re-reviewed by reviewer #2, whose comments you can see below. This reviewer is satisfied that their earlier concerns have been addressed.

Response:

Thank you for this positive update. We are glad that the reviewer’s comments were satisfactorily addressed and believe they improved the manuscript.

Comment:

However, I note that although reviewer #1 requested a copy of the interview guide, it does not appear in your submission. Please could you upload a copy of your interview guide as a supporting information file?

Response:

Thank you for this important observation. We have now attached both the PRE and POST bronchoscopy questionnaires. 

Comment:

Response:

We have included a rebuttal letter in response to the Associate Editor’s comments.

Comment:

Response:

We have not made any changes to the original version. We therefore have not attached a new/another Manuscript.

Comment:

Response:

We have not made any changes to the current version. We therefore have not attached a new/another Manuscript.

Comment:

Response:

We have not made any changes to the financial disclosure statement as it currently stands.

---

## [Editor Report · Decision Letter 3]

9 Oct 2023

Perceptions about and reasons for participation in research bronchoscopy in Uganda: a qualitative analysis

PONE-D-23-01054R3

Dear Dr. Mafigiri,

We’re pleased to inform you that your manuscript has been judged scientifically suitable for publication and will be formally accepted for publication once it meets all outstanding technical requirements.

Kind regards,

Steve Zimmerman, PhD

Associate Editor, PLOS ONE
---

## [Editor Report · Acceptance letter]

11 Oct 2023

PONE-D-23-01054R3 

Perceptions about and reasons for participation in research bronchoscopy in Uganda: a qualitative analysis 

Dear Dr. Kaawa-Mafigiri:

I'm pleased to inform you that your manuscript has been deemed suitable for publication in PLOS ONE. Congratulations! Your manuscript is now with our production department. 

Kind regards, 

on behalf of

Dr Steve Zimmerman 

Staff Editor

PLOS ONE